# Roles of O-GlcNAcylation in Mitochondrial Homeostasis and Cardiovascular Diseases

**DOI:** 10.3390/antiox13050571

**Published:** 2024-05-06

**Authors:** Zhen Qiu, Jiahui Cui, Qin Huang, Biao Qi, Zhongyuan Xia

**Affiliations:** 1Department of Anesthesiology, Renmin Hospital of Wuhan University, Wuhan 430060, China; rm003724@whu.edu.cn (Z.Q.); lw3034880046@163.com (J.C.); huangqin199306@163.com (Q.H.); 2Department of Anesthesiology, Hubei 672 Orthopaedics Hospital of Integrated Chinese and Western Medicine, Wuhan Orthopaedics Hospital of Intergrated Traditional Medicine Chinese and Western Medicine, The Affiliated Hospital of Wuhan Sports University, Wuhan 430070, China

**Keywords:** O-GlcNAcylation, mitochondrial protein, mitochondrial homeostasis, posttranslational modification, cardiovascular diseases

## Abstract

Protein posttranslational modifications are important factors that mediate the fine regulation of signaling molecules. O-linked β-N-acetylglucosamine-modification (O-GlcNAcylation) is a monosaccharide modification on N-acetylglucosamine linked to the hydroxyl terminus of serine and threonine of proteins. O-GlcNAcylation is responsive to cellular stress as a reversible and posttranslational modification of nuclear, mitochondrial and cytoplasmic proteins. Mitochondrial proteins are the main targets of O-GlcNAcylation and O-GlcNAcylation is a key regulator of mitochondrial homeostasis by directly regulating the mitochondrial proteome or protein activity and function. Disruption of O-GlcNAcylation is closely related to mitochondrial dysfunction. More importantly, the O-GlcNAcylation of cardiac proteins has been proven to be protective or harmful to cardiac function. Mitochondrial homeostasis is crucial for cardiac contractile function and myocardial cell metabolism, and the imbalance of mitochondrial homeostasis plays a crucial role in the pathogenesis of cardiovascular diseases (CVDs). In this review, we will focus on the interactions between protein O-GlcNAcylation and mitochondrial homeostasis and provide insights on the role of mitochondrial protein O-GlcNAcylation in CVDs.

## 1. Introduction

In the mid-1980s, Hart and colleagues described the modification of nuclear and cytoplasmic proteins by a single N-acetylglucosamine (GlcNAc) through O-linkage on serine and threonine residues which was named O-linked β-N-acetylglucosamine (O-GlcNAc)-modification (O-GlcNAcylation) [1]. O-GlcNAcylation is a reversible, posttranslational and regulatory modification of proteins in the nucleus, mitochondria and cytoplasm to respond to cellular stress [2]. The level of protein O-GlcNAc in cells is affected by the hexosamine synthesis pathway. When the contents of glucose, glucosamine, and free fatty acids increases, or when the key enzymes in the hexosamine pathway are overexpressed, the end product of hexosamine pathway UDP-GlcNAc is increased to up-regulate the overall O-GlcNAcylation level of intracellular proteins [3,4]. Therefore, O-GlcNAcylation has been proposed as a nutrient sensor and metabolic regulator.

The posttranslational modification of mitochondrial proteins is crucial for regulating mitochondrial homeostasis. Studies have found that O-GlcNAcylation plays an important role in maintaining mitochondrial function and energy metabolism [5,6]. Mitochondrial proteins that are rich in serine and threonine are the main targets of O-GlcNAcylation. A large number of mitochondrial proteins that are O-GlcNAcylated and specifically modified by mitochondrial O-GlcNAc transferase (OGT) have been identified through high-throughput proteomics under physiological and pathological conditions, and the impact of these modifications on global perturbations of mitochondrial function has also been studied [7,8,9]. The adaptive regulation of energy disturbances in the O-GlcNAc cycle has been shown to be crucial for maintaining mitochondrial homeostasis [5,6]. The imbalance of O-GlcNAcylation in mitochondria is related to various mitochondrial dysfunctions, such as abnormal mitochondrial dynamics, decreased mitochondrial biosynthesis, disruption of electron transport chains (ECTs), oxidative stress and the calcium paradox and mitophagy, as well as the activation of mitochondrial apoptosis pathways [10,11,12,13].

As the main energy-demanding organ, the heart largely relies on ATP produced by mitochondrial energy metabolism to promote contractile function and myocardial cell metabolism [14,15]. Integral mitochondrial homeostasis is crucial for cardiac contractile function and myocardial cell metabolism, and the imbalance in mitochondrial homeostasis plays a crucial role in the pathogenesis of cardiovascular diseases (CVDs) [16], mainly by regulating cellular energy metabolism [17], reactive oxygen species (ROS) and autophagy [18], intracellular calcium level [19], apoptotic [20] and other cell- or tissue-specific mechanisms [21]. What is more, the O-GlcNAcylation of cardiovascular proteins is a dynamic process and crucial for maintaining myocardial cell function. Elevated O-GlcNAcylation in the cardiovascular system is considered as a potential self-protection warning signal or pressure response [22,23]. And enhancing O-GlcNAcylation can increase the survival rate of cells under acute stress conditions, including hypoxia, local ischemia, oxidative stress and so on [22,24,25]; both decreased and increased O-GlcNAc levels can promote heart failure [26]. All these findings indicate that the imbalance of O-GlcNAcylation is closely related to CVDs. Therefore, we will review the mitochondrial homeostasis regulated by protein O-GlcNAcylation and the key roles of O-GlcNAcylation in the pathogenesis of CVDs, including heart failure, cardiac hypertrophy, myocardial infarction (MI) and myocardial ischemia/reperfusion (MI/R) injury.

## 2. O-Linked β-N-Acetylglucosamine-Modification (O-GlcNAcylation)

O-GlcNAcylation is a monosaccharide modification that occurs on acetylglucosamine linked to the hydroxyl terminus of serine and threonine in proteins. O-GlcNAc dynamically modifies nuclear, cytoplasmic and mitochondrial proteins through the hexosamine biosynthesis pathway (HBP) after translation [27]. The glycosylation level of the intracellular protein O-GlcNAc is influenced by HBP. Multiple nutrients contribute to both UDP-GlcNAc and protein O-GlcNAcylation. Glucose availability is an important factor in O-GlcNAc synthesis, and glutamine is critical as the amine donor for glucosamine 6-phosphate, while fatty acid metabolism is probably the main source for the acetyl moiety. When the contents of glucose, glucosamine or free fatty acids increases, or when the key enzymes of HBP are overexpressed, UDP-GlcNAc correspondingly increases, leading to an increase in the overall O-GlcNAcylation level of intracellular proteins. O-GlcNAcylation is promoted by two enzymes with opposite functions. OGT covalently connects the nucleotide sugar UDP-GlcNAc produced from HBP to proteins. O-GlcNAcase (OGA) hydrolyses GlcNAc from proteins, regulates key protein functions, such as protein stability, degradation or function, and regulates protein activity in dynamic cycles for precise regulation [28,29]. To date, more than 4000 proteins with O-GlcNAc linkages on serine and threonine residues have been identified [30]. O-GlcNAcylation senses the levels of extracellular glucose, glutamine, fatty acids and other energy molecules and competes with phosphorylation modifications for binding sites, changing the direction of signal transmission to complete signal termination or continuation to affect important biological processes such as transcription, translation, metabolism and the cell cycle (Figure 1).

### The Modulation of O-GlcNAcylation and Its Therapeutic Implications

The recycling of protein O-GlcNAcylation is mediated by two enzymes: OGT and OGA, which catalyze the addition and removal of O-GlcNAc, respectively [31]. On the other hand, UDP-GlcNAc is the final product of the HBP, which is mainly regulated by glucose-6-phosphate glutamine: fructose-6-phosphoamidotransferase (GFAT). The inhibition of GFAT activity results in decreased protein O-GlcNAcylation [32]. Therefore, by determining the abundance of the donor substrate UDP-GlcNAc and regulating the activity of OGT, OGA and GFAT enzymes, as well as the expression levels of their respective adapter proteins and substrates, the cellular O-GlcNAcylation level is regulated [28,29,33].

These three key enzymes are potential targets for therapeutics. Glucosamine or glutamine treatment promotes the activity of O-GlcNAcylation by increasing HBP flow. Pharmacological inhibitors of OGA include PUGNAc [5,34,35], Thiamet G [36,37], NAG thiazoline [38] or NButG [8,39], as well as the genetic inhibition of O-GlcNAcase by siRNA, or overexpressing OGT expression through adenovirus could increase the formation of O-GlcNAc [26]. The pharmaceutical GFAT inhibition with the glutamine analogue azaserine, or GFAT inhibitor DON [40] and Azaserine [41], or OGT inhibition with TTO4, OSMI or UDP-5SGlcNAc [42,43,44] can block O-GlcNAcylation. However, none of the enzyme blockers are entirely specific. Alloxan has inhibitory effects on both OGT and OGA [45]. O-GlcNAcylation can also be blocked by the overexpression of OGA by adenovirus, inhibition of OGT by OGT siRNA or gene-editing techniques for OGT gene blockade [34,46,47].

Most O-GlcNAcylated proteins are located in the nucleus, cytoplasm and mitochondria. Mitochondria are the main site of ATP production in eukaryotes, and mitochondrial proteins are rich in serine and threonine, making them the main targets of O-GlcNAc [5]. O-GlcNAcylation of the mitochondrial proteome may regulate mitochondrial energy metabolism by regulating the tricarboxylic acid cycle (TCA) and the composition of the mitochondrial respiratory chain, thereby controlling oxygen consumption and ATP production [48]; it also involves the threshold of calcium-induced permeability transition pore opening, mitochondrial dynamics, mitochondrial morphology, acetylation of mitochondrial proteomes and mitochondrial oxidative stress [49]. A study confirmed that both OGT and OGA are located in mitochondria [50]. OGT catalyzes the O-GlcNAcylation of mitochondrial oxidative phosphorylation proteins, respiratory chain complexes (I, III, VI), dynamic-related protein 1 (DRP1), voltage-dependent anion channel (VDAC), calcium/calmodulin-dependent protein kinase II (CaMKⅡ) and the autophagy protein SNAP-29, thereby regulating the dynamic balance of mitochondrial energy metabolism, kinetics, oxidative stress and autophagy to maintain mitochondrial homeostasis [13,49,51,52]. Ee Phie et al. reported that O-GlcNAcylation was increased by inhibiting OGA or adding GlcN, thereby reprogramming mitochondrial function and altering energy metabolism [5]. These data collectively demonstrate the unique role of O-GlcNAcylation in regulating mitochondrial function and homeostasis.

Protein O-GlcNAcylation plays a key role in maintaining normal cell function. An imbalance in O-GlcNAcylation homeostasis is involved in the pathogenesis of multiple diseases, especially chronic diseases such as diabetes and CVDs [53]. The acute increase in O-GlcNAcylation seems to protect against stress-induced cardiovascular injury [22]. An increase in O-GlcNAc levels during hypoxia or reperfusion can improve functional recovery and exert anti-ischemic myocardial protection through various mechanisms [38,54]. However, excessive increased O-GlcNAcylation leads to cardiomyocyte hypertrophy, fibrosis and diastolic dysfunction [14,55]. O-GlcNAcylation levels are increased in the cardiac tissue of patients who undergo aortic valve stenosis surgery and rat models of hypertension [56]. Both decreased and/or increased O-GlcNAcylation can promote heart failure, and maintaining O-GlcNAcylation homeostasis is essential for normal myocardial cell function, as shown by using patient samples and mice model of heart failure [57,58]. In summary, all of these findings suggest that the O-GlcNAcylation of cardiovascular proteins is a dynamic process that is crucial for maintaining normal myocardial cell function. Another factor influencing the different responses to changes in O-GlcNAcylation levels is the duration and degree of O-GlcNAcylation upregulation.

## 3. O-GlcNAcylation and Mitochondrial Homeostasis

### 3.1. O-GlcNAcylation of Mitochondrial Proteins

Under normal conditions, the morphology, structure and function of mitochondria are maintained in a relatively stable dynamic equilibrium, known as mitochondrial homeostasis; this state is dynamically regulated by pathways such as mitochondrial fusion and division (maintaining mitochondrial structural integrity), biosynthesis and autophagy (maintaining a “healthy” mitochondrial state) and oxidative stress (regulating mitochondrial energy metabolism) [59]. Mitochondria are the main sites of life for eukaryotic cells, and they are crucial for maintaining and regulating the metabolic functions of cells and organs. An imbalance in mitochondrial homeostasis is associated with many diseases, such as CVDs, diabetes, metabolic syndrome, neurodegeneration, cancer and ageing [9,60]. There are various reversible posttranslational protein modifications in mitochondria that play a role in mediating various mitochondrial functions. With the emergence of large-scale proteomic screening techniques, it has been found that the array of known mitochondrial posttranslational modifications and their protein targets has increased significantly [61]. O-GlcNAcylation exists widely and dynamically in mitochondria, and mitochondrial proteins are the main targets of O-GlcNAc which is the key modulator of mitochondrial function and energy metabolism [62].

OGT and OGA, as the key enzymes that regulate the balance of O-GlcNAcylation, exist in mitochondria. OGT or OGA overexpression significantly decreases the expression of mitochondrial proteins involved in the respiratory chain and the TCA cycle, which indicates that the O-GlcNAcylation cycle can occur directly in mitochondria and the changes in the O-GlcNAcylation cycle can profoundly affect mitochondrial energy metabolism [63]. There are three subtypes of OGT: short OGT (sOGT); mitochondrial OGT (mOGT); and nucleocytoplasmic OGT (ncOGT) [64]. mOGT is the main isoform of mitochondrial O-GlcNAcylated protein. Sacoman et al. used high-throughput proteomics to identify 84 mitochondrial glycoproteins in HeLa cells and reported that the downregulation of endogenous mOGT could change the mitochondrial structure and function through a mechanism involving Drp1-dependent mitochondrial rupture, decreased mitochondrial membrane potential (MMP) and the loss of mitochondrial ROS (mROS). The decreased expression of ncOGT and mOGT subtypes is associated with increased mitochondrial respiration and glycolysis, while ncOGT mainly regulates cellular bioenergetics [65]. In another human study, analysis of the genome sequence showed that overexpression of the ncOGT subtype resulted in an increased O-GlcNAcylation of mitochondrial proteins, indicating that ncOGT is necessary and sufficient for the production of O-GlcNAc-modified mitochondrial proteomes [66]. Further research revealed that mOGT expression is glucose dependent. The upregulation of mOGT expression leads to decreased cellular ATP levels, affecting the mitochondrial transmembrane potential and increasing the production of mitochondrial ROS. Many mitochondrial proteins have been identified as mOGT substrates [9]. These proteins are mostly located in the mitochondrial matrix and inner membrane, participating in mitochondrial respiration, fatty acid metabolism, transportation, translation, apoptosis and mitochondrial DNA (mtDNA) processing [67,68]. These findings indicate that mOGT interacts with and modifies many mitochondrial proteins and that its dysregulation can affect cellular bioenergetics and mitochondrial function. Moreover, short OGA co-transfection significantly reduced the O-GlcNAcylation of the biosensor, and the overexpression of short OGA increased the mROS levels [50]. These findings indicate that the short OGA subtype targets mitochondria, thereby regulating ROS homeostasis.

With the discovery of enzymes and substrates of the O-GlcNAc cycle in mitochondria, the adaptive regulation of energy disturbances by the O-GlcNAc cycle has been shown to be crucial for maintaining mitochondrial homeostasis. A large number of O-GlcNAcylation mitochondrial proteins and mOGT specific modified proteins have been identified through high-throughput proteomics under physiological and pathological conditions, and the impact of these modifications on global perturbations of mitochondrial function has also been studied. A study identified 409 O-GlcNAcylated mitochondrial proteins in the O-GlcNAcylome of isolated cardiac mitochondria in rats, and among them, 191 mitochondrial proteins have increased O-GlcNAcylation in response to NButGT. This is related to the enhanced activity of complex I (CI) with increased maximal respiration in the presence of pyruvate–malate, and significantly reduced mitochondrial ROS release, which may be related to the O-GlcNAcylation of specific subunits of the ETC complexes (CI, CIII) and TCA cycle enzymes [8]. These results indicated that the dynamic mitochondrial O-GlcNAcylation system could rapidly modify mitochondrial function. In another rat study, mitochondrial proteins of liver were obviously O-GlcNAcylated according to immunoblot profiling, and 14 O-GlcNAcylation sites were identified among the O-GlcNAcylated mitochondrial proteins. All of these O-GlcNAcylated mitochondrial proteins are enzymes which are involved in various biological processes such as the urea cycle, TCA cycle and lipid metabolism [69]. Ma J also found 88 mitochondrial proteins of the heart from rats to be O-GlcNAcylated, and the oxidative phosphorylation system was a major target. Meanwhile, TMG treatment resulted in a significant increase in the mitochondrial oxygen consumption rate, ATP production rate and an enhanced threshold for PTP to open by Ca^2+^ [70]. Gu et al. reported that numerous mitochondrial proteins in C2C12 mice myoblast cells are O-GlcNAcylated under basal conditions but are altered under high-glucose conditions [71]. All of these findings suggest an important role of O-GlcNAcylation in regulating mitochondrial protein activation and mitochondrial function.

### 3.2. The Regulatory Mechanisms of Protein O-GlcNAcylation on Mitochondrial Homeostasis

Mitochondrial protein O-GlcNAcylation plays an important role in regulating cellular function; the destruction of O-GlcNAcylation homeostasis is involved in the pathogenesis of chronic diseases such as diabetes [43,72,73], neurodegenerative diseases [74,75], tumors [28,76,77] and CVDs [14,26,44]. Insulin resistance and diabetic CVDs are promoted by O-GlcNAcylation through an increase in the imbalance of multiple signaling pathways at the transcriptional, translational and posttranslational levels. Ma et al. found that 86 mitochondrial proteins were O-GlcNAcylated in streptozotocin (STZ)-induced diabetic rat cardiac tissue. Many proteins exhibit site-specific changes in O-GlcNAcylation in response to diabetes, which indicates that protein O-GlcNAcylation is a new regulatory function that mediates adaptive changes in mitochondrial metabolism during the progression of diabetic cardiomyopathy [78]. Banerjee et al. reported that STZ-treated diabetic rats exhibited an increase in mitochondrial OGT and a concomitant decrease in mito-specific OGA. The inhibition of OGT or OGA activity significantly affects energy production, the MMP and mitochondrial oxygen consumption [79]. These data indicate that cardiac mitochondria not only have robust O-GlcNAc circulation, but also that O-GlcNAcylation dysregulation plays a key role in diabetes-related mitochondrial dysfunction.

#### 3.2.1. Mitochondrial Dynamics (Fusion–Fission)

Mitochondria are very dynamic organelles. A single mitochondrion can have multiple forms which are controlled by mitochondrial fusion and fission. Another dynamic aspect of mitochondria is the selective removal of dysfunctional mitochondria to ensure healthy mitochondrial quality. These mitochondrial dynamics include fusion, fission, selective degradation and transport processes [80]. The key components that mediate mitochondrial fusion and fission are members of the GTP enzyme dynamic protein family, which utilize GTP hydrolysis to drive the mechanical activity of biological membranes. Dynamic-related proteins (DRPs) are multidomain GTPases that reshape membranes and mediate the dynamic fusion and division of mitochondria through GTPase-stimulated self-assembly and GTP hydrolysis-mediated conformational changes. In mammals, three DRP families, namely, mitochondrial fusion protein (Mfn1/2), optic atrophy 1 (OPA1) and Drp1, are crucial for mitochondrial fusion and division. Metabolomic analysis revealed that drug treatment reduced the protein O-GlcNAcylation level of M2 macrophages in breast tumors, which manifested as a change in mitochondrial fragmentation [81]. Recent research has demonstrated that under hyper O-GlcNAcylation conditions caused by OGA deficiency, the size of the mitochondria significantly decreases, while the quantity and total mass of mitochondria correspondingly increase. Due to the increase in mitochondrial content, OGA knockout cells exhibited comparable coupled mitochondrial OXPHOS and ATP levels, with an increased content of O-GlcNAcylation of Drp1. Other modification sites are predicted to be in the GTPase domain and affect the activity of the enzyme [73]. OGT activation reduces mitochondrial motility. The mitochondrial motor-adaptor protein Milton is an essential substrate for OGT that prevents mitochondrial motility by locating and mutating key O-GlcNAcylated serine residues. By dynamically adjusting Milton O-GlcNAcylation, OGT adjusts mitochondrial dynamics based on nutrient availability [82]. These results emphasize the important role of O-GlcNAcylation in mitochondrial dynamics.

Drp1 is the central mediator of mitochondrial fission, and the active translocation of Drp1 from the cytoplasm to the mitochondria drives the division of the outer membrane of mitochondria. Drp1 is O-GlcNAcylated at threonine 585 and 586 in cardiomyocytes. O-GlcNAcylation was significantly enhanced by the chemical inhibition of N-acetyl-glucosaminidase. An increased level of O-GlcNAcylation could upregulate the level of Drp1 and induce the translocation of Drp1 from the cytoplasm to the mitochondria. Mitochondrial fragmentation and a decreased MMP are also accompanied by increased O-GlcNAcylation [83]. OPA1 can play a specific role in controlling IMM fusion and ridge structures based on certain protein modifications. Makino’s study revealed that high-glucose conditions led to a decreased OPA1 protein level with increased O-GlcNAcylation, which led to mitochondrial dysfunction by increasing mitochondrial fragmentation, reducing the MMP and attenuating the activity of mitochondrial complex IV [84].

When acutely or chronically induced, Drp1 O-GlcNAcylation can affect the balance of mitochondrial fusion and fission differently. Park et al. reported that Aβ regulates mitochondrial fission by increasing the O-GlcNAcylation of Drp1 in the AD mouse brain [75]. During cerebral ischemia/reperfusion (I/R) injury, OGT knockdown markedly decreased the level of phosphorylated Drp1 at serine 637 and promoted Drp1 translocation from the cytosol to the mitochondria to increase the neurological score and infarct volume. The specific Drp1 inhibitor mdivi-1 effectively alleviated I/R-induced brain injury in OGT knockdown mice [85]. O-GlcNAcylation could regulate Drp1-mediated mitochondrial fission and apoptosis under high glucose in a diabetic retinopathy rat model [13]. These findings suggest that the balance of mitochondrial fission and fusion is regulated by O-GlcNAcylation of GTPases in response to cellular stimulation.

#### 3.2.2. Mitochondrial Biosynthesis

Mitochondrial biosynthesis is also an important way to regulate mitochondrial homeostasis. Mitochondrial biosynthesis is the process of generating new mitochondria from existing mitochondria and requires cooperation between the mitochondrial genome and nuclear genome. It is controlled by both mtDNA and nuclear DNA, and involves the replication of mtDNA and the synthesis and transportation of mitochondrial proteins.

The PGC1-α/NRFs pathway is a central link in regulating mitochondrial biosynthesis. Internal or external stimuli such as oxidative stress, inflammation and mitochondrial drug toxicity can be regulated by the PGC1-α/NRFs pathway to affect mtDNA replication and the synthesis of nuclear encoded mitochondrial proteins, thereby mediating mitochondrial biogenesis. Housley first identified the novel posttranslational modification O-GlcNAcylation of PGC1-α. PGC1-α binds to OGT and targets the enzyme to FoxOs, leading to an increase in its O-GlcNAcylation and transcriptional activity. Glucose enhances the activation of FoxO1 via PGC1-α/OGT-mediated O-GlcNAcylation. Therefore, as a co-activator of transcription, PGC-1α is important for nutrient/stress sensing and energy metabolism by targeting OGT to increase the O-GlcNAcylation of specific transcription factors [86]. Ruan et al. reported that glucose availability modulates gluconeogenesis by regulating PGC-1α O-GlcNAcylation and stability via the OGT/HCF-1 complex [87]. Wang et al. found that OGA can modulate mitochondrial density by PGC-1α and mitochondrial function through protein O-GlcNAcylation [88]. Furthermore, OGT deletion leads to decreased mitochondrial protein expression with decreased PGC1-α protein expression, and O-GlcNAcylation is essential for cold-induced thermogenesis and mitochondrial biogenesis in brown adipose tissue [89].

Tan reported that elevated levels of O-GlcNAc lead to mitochondrial elongation and MMP upregulation. The sustained O-GlcNAcylation in the mouse brain and liver confirmed the metabolic phenotypes observed in these cells, and the knockdown of OGT increased ROS levels and the antioxidant response of nuclear factor erythroid 2-related factor 2 (NRF2) with impaired respiration [5]. Chen et al. identified that KEAP1, the main negative regulator of NRF2, as a direct substrate of OGT and O-GlcNAcylation of KEAP1 at serine 104 is necessary for the efficient ubiquitination and degradation of NRF2 [90]. Augmented O-GlcNAc signaling also increased the protein expression levels of phospho-Akt, phospho-GSK-3β, NRF2 and Bcl-2, and decreased the levels of Bax and cleaved caspase-3 [91]. These suggested the biochemical and functional connections between O-GlcNAcylation and NRF2, jointly demonstrating the unique role of O-GlcNAcylation in regulating mitochondrial function.

#### 3.2.3. Mitochondrial Oxidative Stress

Mitochondria are the main sites for ROS production in cells. Under normal conditions, intracellular ROS can act as intracellular signaling molecules to regulate normal physiological functions of the body. When tissues are damaged, mROS increase significantly, which damages mitochondria and leads to mitochondrial dysfunction.

Pagesy et al. reported that, in HEK-293T cells and RAW 264.7 cells, the co-transfection of short OGA significantly reduced the O-GlcNAcylation of the biosensor, and the overexpression of short OGA increased mitochondrial ROS levels, indicating that the short OGA targets mitochondria, thereby regulating ROS homeostasis [50]. OSMI-1 significantly increased ROS-induced endoplasmic reticulum (ER) stress, resulting in a decrease in Bcl2 and the release of cytochrome c from mitochondria [92]. Deletion of the OGT gene aggravated the homeostatic imbalance and oxidative stress in mitochondria induced by cold stress in skeletal muscle [93]. Jóźwiak et al. reported that the increased expression of mOGT affects the mitochondrial transmembrane potential and increases the production of mROS. Thus, it participates in mitochondrial respiration, apoptosis and mtDNA processes, thereby affecting cellular bioenergy and mitochondrial function [9]. Increased oxidative stress leading to mtDNA damage in cardiomyocytes is an important factor in the pathogenesis of diabetes cardiomyopathy. In the diabetic mouse heart, researchers have observed an increased mtDNA damage caused by the hyperglycemia-induced O-GlcNAcylation of Ogg1 to mediate mitochondrial oxidative stress damage [94]. Ngoh et al. discovered that O-GlcNAcylation plays a protective role in the mouse heart by attenuating the formation of the mitochondrial permeability transition pore (mPTP) and the subsequent loss of MMP. Mitochondrial dysfunction and ROS production can promote chronic diseases. OGT or OGA overexpression disrupt protein posttranslational modification via O-GlcNAcylation, impairing mitochondrial function. Sustained O-GlcNAc elevation increases OGA expression and reduces cellular respiration and ROS generation [5]. Other research studies have shown that deregulated hyper-O-GlcNAcylation promotes the progression of NAFLD by reducing mitochondrial oxidation and promoting hepatic lipid accumulation [95]. Additionally, increased O-GlcNAcylation could resist oxidative stress and promote mitochondrial respiration in the retina of aged rats [96].

Calcium ions enter and leave mitochondria in a tissue-specific manner through various unique channels and transport proteins. Mitochondria have a unique ability to accumulate large amounts of calcium. Mitochondrial calcium is a double-edged sword. Low levels of calcium are crucial for maintaining the optimal rate of ATP production, but extreme levels of calcium leads to the loss of mitochondrial function. During cellular stress, mitochondrial calcium overload induces the loss of MMP and a burst of ROS. Research has found that AdOGT and OGA inhibition could significantly reduce the calcium overload and ROS production induced by hypoxia and oxidative stress, which may be a mechanism by which O-GlcNAcylation reduces ischemia/hypoxia-mediated mPTP formation in neonatal rat cardiac myocytes or mice [97]. One study reported that enhanced O-GlcNAcylation in the heart can lead to impaired calcium dynamics and contractile derangements, arrhythmias associated with voltage-gated sodium channels and CaMKII activation, as well as mitochondrial dysfunction [44]. Yoon et al. reported that PUGNAc enhances O-GlcNAc-mediated augmentation to increase human corneal endothelial cells’ viability, decrease the loss of ΔΨ(m) and inhibit intracellular ROS in the context of tBHP-induced oxidative stress and mitochondrial calcium overload [98]. It has been reported that enhanced mitochondrial Ca^2+^ is able to increase CaMKII activity. Bers et al. showed that CaMKII is highly O-GlcNAcylated in the hearts and brains of patients with diabetes; an acute increase in the extracellular glucose concentration leads to O-GlcNAcylation of CaMKII, which results in the release of Ca^2+^ from the SR and induces ROS production [6]. TMG treatment of rats significantly increased mitochondrial oxygen consumption rates, ATP production rates and raised the threshold Ca^2+^ for mPTP opening [70].

#### 3.2.4. Mitophagy

In order to maintain mitochondrial and cellular homeostasis and prevent damaged mitochondria from damaging cells, cells selectively encapsulate and degrade damaged or dysfunctional mitochondria within the cell, known as mitophagy. Mitophagy, as an important mitochondrial quality control mechanism, can lead to the gradual accumulation of mtDNA mutations under conditions of stress, such as ROS stress, as well as a decrease in the intracellular MMP and depolarization damage, ultimately leading to cell death [99]. Timely elimination of damaged mitochondria is a self-protective mechanism of cells. The disruption of mitophagy can trigger various diseases. Li et al. showed that the inhibition of cell proliferation in OGT-deficient cells of mice is due to secondary mitochondrial dysfunction caused by the excessive activation of mTOR. In normal cells, OGT maintains a low mTOR activity and mitochondrial fitness by suppressing proteasome activity [12].

Ubiquitin-dependent and ubiquitin-independent pathways are two methods of mitophagy. The ubiquitin-dependent pathway relies on the widespread ubiquitination of mitochondrial surface proteins to promote mitophagy. The PINK1/Parkin pathway is currently the most extensively studied pathway. PINK1 interacts with Parkin to jointly regulate the mitophagy process to maintain mitochondrial quality [100]. Recently, research has found that mitochondrial homeostasis was affected by O-GlcNAcylation through regulating Parkin [101]. Murakami et al. revealed that OGT ensures mitochondrial quality through PINK1-dependent mitophagy, thereby strictly regulating maintenance and stress responses [102]. Enhanced O-GlcNAcylation may mediate GSK-3β inactivation, an increased p-Drp-1/MFN2 ratio and Smad2 activation to promote hypertrophy, mitophagy and fibrosis, thereby exacerbating IH4W-induced mice right ventricular dysfunction and remodeling [55]. A decreased level of O-GlcNAcylation enhances autophagy flux by promoting the fusion of autophagosomes and lysosomes. Inhibiting OGT and reducing O-GlcNAcylation levels can reduce mTOR activation and promote mTOR-dependent autophagy in rat cortical neurons [103]. Furthermore, CaMKII mediates OGT phosphorylation to induce O-GlcNAcylation and the activation of ULK, thereby mediating hunger-induced mouse liver autophagy to maintain systemic homeostasis [104]. Cellular autophagic flux is significantly increased due to the inhibition of O-GlcNAcylation, whereas it decreases at high levels of O-GlcNAcylation. AMPK O-GlcNAcylation suppressed the activity of this regulator to inhibit the activity of ULK1 and autophagy [105]. On the other hand, OGT could regulate autophagy by mediating the O-GlcNAcylation of SNAP-29 protein in mammalian cells and in C. elegans [106]. Thus, O-GlcNAcylation regulates mitophagy by recycling intracellular energy or removing cytotoxic proteins to maintain mitochondrial homeostasis.

#### 3.2.5. Mitochondrial Apoptosis

The pathways involved in cell apoptosis can be divided into the endogenous mitochondrial pathway, endogenous endoplasmic reticulum pathway and exogenous death receptor pathway. When the MMP decreases, proapoptotic factors are released into the cytoplasm and form apoptotic complexes with the assistance of ATP and dATP, which recruit and activate pro-caspase-9 to form the caspase-9 holozyme, initiating the caspase cascade reaction and leading to cell apoptosis [107]. Proteomic analysis identified VDAC, which is an important modulator of mitochondria-related apoptosis, as a potential target of O-GlcNAcylation. Mitochondria with elevated O-GlcNAc levels have high levels of VDAC O-GlcNAcylation and are more resistant to calcium-induced swelling in mouse hearts [108]. Palaniappan et al. identified a functional connection between O-GlcNAcylation and mitochondrial apoptosis through VDAC [52]. Further research indicated that IFIT3 regulates the O-GlcNAcylation of VDAC2 by stabilizing the interaction of VDAC2 with OGT, and the increased O-GlcNAcylation of VDAC2 protects pancreatic ductal adenocarcinoma cells from chemotherapy-induced apoptosis by using the highly metastatic pancreatic ductal adenocarcinoma cell line L3.6pl and patient-derived primary cell TBO368 [109].

The decrease in O-GlcNAcylation occurred synchronously with the loss of MMP in neonatal rat cardiac myocytes; and the enhancement of O-GlcNAc levels attenuated the loss of MMP. A study revealed that siRNA or an inhibitor targeting OGA significantly increases O-GlcNAcylation levels and reduces neonatal rat cardiac myocytes apoptosis after hypoxia. The inhibition of OGA activity restores MMP after hypoxia, while the enhancement of its activity impairs the restoration of MMP [110]. OGT inhibition significantly reduced O-GlcNAcylation, exacerbated myocardial cell apoptosis after hypoxia and sensitized myocytes to MMP collapse. The inhibition of OGT reduces the O-GlcNAcylation level of VDAC in isolated mitochondria and sensitizes cells to calcium-induced mPTP formation, indicating that mPTPs may be an important target of O-GlcNAcylation signaling [111]. Beclin-1 and anti-apoptotic Bcl-2, which play critical early roles in autophagy, are targets of O-GlcNAcylation [112]. Both glucosamine and OGT overexpression could increase basal and I/R-induced O-GlcNAcylation levels and significantly attenuated the loss of MMP by increasing mitochondrial Bcl-2 levels to decrease cell injury. However, the downregulation of OGT leads to a decrease in basal O-GlcNAc levels, prevents the I/R-induced increases in O-GlcNAcylation and mitochondrial Bcl-2 and exacerbates neonatal rat cardiomyocytes injury [113].

Therefore, O-GlcNAcylation is closely related to mitochondrial apoptosis, and it can play an important role in mitochondria-mediated apoptosis by regulating the O-GlcNAcylation activity of VDAC, the activation of the mitochondrial Bcl-2 level, mPTPs and MMP.

#### 3.2.6. Mitochondrial Energy Metabolism

It has been reported that O-GlcNAcylated peptides exist in mitochondria which are involved in energy transduction [114]. Immunoprecipitation in neonatal rat cardiomyocytes and AC16 (human hybrid) cells revealed that several mitochondrial proteins, which are members of mitochondrial respiratory chain complexes, such as the NDUFA9 subunit of complex I, the core 1 and core 2 subunits of complex III and the mtDNA-encoded subunit I of complex IV, are O-GlcNAcylated. These O-GlcNAcylated mitochondrial proteins are enzymes involved in various biological processes, such as urea cycle, TCA cycle and lipid metabolism, indicating the important role of protein O-GlcNAcylation in mitochondrial function [8,69,115,116]. Increased mitochondrial O-GlcNAcylation is associated with the impaired activity of complexes I, III and IV, as well as decreased mitochondrial calcium and cellular ATP content. When O-GlcNAcylation is decreased, mitochondrial function improves, complex I, III and IV activity increases to normal levels and mitochondrial calcium and cellular ATP levels return to control levels [115,116]. OGT or OGA overexpression in mice significantly decreased the expression of mitochondrial proteins involved in the respiratory chain and TCA cycle, resulting in changes in mitochondrial morphology, also reducing cellular respiration and glycolysis. These findings indicate that changes in the O-GlcNAc cycle can profoundly affect mitochondrial energy metabolism [63,117]. OGA deficiency in mouse embryonic fibroblasts leads to hyper O-GlcNAcylation, with a significant decrease in mitochondrial size, a corresponding increase in quantity and total mitochondrial mass and exhibiting comparable levels of mitochondrial Oxphos and ATP coupling [73]. These findings indicate that O-GlcNAcylation is crucial for the normal energy function of mitochondria [118].

OGT is a nutrient sensor sensitive to glucose flux. The decreased expression of sncOGT and mOGT is related to increased mitochondrial respiration and glycolysis, indicating that ncOGT is a regulatory factor for mitochondrial biogenesis and metabolism. Four proteins related to mitochondrial biogenesis and metabolism regulation, including leucine-rich PPR protein and mitochondrial aconitic acid hydratase, were identified as candidate substrates of mOGT in Hela cells [65]. High OGT activity is essential for maintaining mitochondrial respiration [67,68]. Sacoman et al. reported that reducing endogenous mOGT expression leads to changes in mitochondrial structure and function, as indicated by changes in Drp1-dependent mitochondrial fragmentation, and decreases MMP [65]. Increased OGT activity induces the dynamic O-GlcNAacylation of the regulatory domain of the glucose rate-limiting enzyme hexokinase 1 (HK1), which subsequently promotes the assembly of the glycolytic metabolites on the outer membrane of mitochondria. This modification enhances the mitochondrial binding of HK1, coordinating glycolysis and mitochondrial ATP production. Mutations in the O-GlcNAcylation site of HK1 reduce ATP production [10]. He et al. reported that the IDH2 protein was highly expressed in colorectal cancer tissues (CRC) and correlated with poor survival in CRC patients. The overexpression of isocitrate dehydrogenase 2 (IDH2) significantly increased glycolysis and TCA cycle metabolites. O-GlcNAcylation enhanced the protein half-time of IDH2 by inhibiting ubiquitin-mediated proteasome degradation [119].

Therefore, the imbalance of O-GlcNAcylation in mitochondria is related to various mitochondrial dysfunctions, such as abnormal mitochondrial dynamics, reduced mitochondrial biosynthesis, disruption of ECT, oxidative stress and the calcium paradox, as well as the activation of mitochondrial apoptosis pathways [51]. The role of O-GlcNAcylation cycle in regulating mitochondrial homeostasis seems complex (Table 1 and Figure 2).

## 4. Role of O-GlcNAcylation in Cardiovascular Diseases (CVDs)

The cardiovascular system is complex, and the posttranslational modification of serine and threonine residues in nuclear, cytoplasmic and mitochondrial proteins by O-GlcNAcylation is considered an important regulatory mechanism in the cardiovascular system. Changes in O-GlcNAcylation may lead to many metabolic imbalances and affect cardiovascular function; and this process has an prominent role in heart remodeling [55,120,121], heart failure [22,26,122], diabetic cardiomyopathy [123,124,125], ischemic heart disease [24,42,126], hypertension [127] and arrhythmia [128,129]. Protein O-GlcNAcylation reprograms the cardiac substrate metabolism under chronic stress to advantageously alter adaptation [130]. The rapid increase in protein O-GlcNAcylation has cardioprotective effects. Dysfunctional cardiomyocyte-specific OGT-KO hearts are more fibrotic and hypertrophic, and glycolytic genes of cardiac are upregulated and lead to progressive cardiomyopathy [131]. Therefore, OGT in myocardial cells is essential for the maturation of mammalian hearts, and the induced loss of cardiomyocyte-specific OGT in adult mice can lead to progressive ventricular dysfunction. The rapidly increasing O-GlcNAcylation in the heart is an adaptive cardiac response that can protect cardiac function [29,132,133]. However, if the level of O-GlcNAcylation remains high in chronic disease or in the long-term (related to diabetes and overnutrition), it will lead to cardiac dysfunction [72,134,135].

### 4.1. Cardiac Hypertrophy and Heart Failure

Heart failure is a leading cause of death worldwide and is related to an increase in the incidence of obesity, hypertension and diabetes [22,136]. It is speculated that pressure-overload hypertrophy (POH) augments HBP flux to increase the O-GlcNAcylation level. The O-GlcNAcylation level of proteins varies during POH, with elevations occurring during active hypertrophic growth early after transverse aortic constriction (TAC) in C57/Bl6 mice [127]. O-GlcNAcylation plays a role in the high-glucose-induced myocardial hypertrophy of neonatal rat cardiomyocytes via ERK1/2 and cyclin D2 [34]. Heart failure has been associated with an increase in the O-GlcNAcylation rate, which induces cardiomyopathy, suggesting a possible role for O-GlcNAcylation in the development of chronic heart dysfunction [14,44,57,121,137,138,139]. Heart failure (regardless of diabetes) both experimentally and clinically is accompanied by marked increases in O-GlcNAcylation. Heightened O-GlcNAcylation in the heart leads to impaired calcium kinetics and contractile derangements, arrhythmias, mitochondrial dysfunction, maladaptive hypertrophy, microvascular dysfunction, fibrosis and cardiomyopathy [63]. Twenty-four-week increases in protein O-GlcNAcylation led to cardiac hypertrophy, mitochondrial dysfunction, fibrosis and diastolic dysfunction by using a naturally occurring dominant-negative dnOGA cardiomyocyte-specific overexpression mouse model [14]. Further research revealed that a partial site of O-GlcNAcylation was observed after 2 weeks of intermittent hypoxia exposure [121]. In cardiomyocyte-specific OGT deletion mice, LV diastolic and systolic volumes were elevated [44]. In mouse TAC hearts, OGT KO significantly reduced O-GlcNAc levels and caused a further decline in LV systolic function [137]. Lunde et al. tested the left ventricular tissue from aortic stenosis patients and rat models of hypertension, myocardial infarction and aortic banding, with and without failure, and found that the overall O-GlcNAcylation was increased by 65% in aortic stenosis, by 47% in hypertensive rats, by 81% and 58% in hypertrophic and failing hearts after aortic banding and by 37% and 60% in hypertrophic and failing hearts after myocardial infarction, respectively. The protein O-GlcNAcylation pattern is different in hypertrophic and failing hearts. The OGT, OGA and GFAT2 protein and/or mRNA levels were increased by pressure overload. Pharmacological inhibition of OGA decreased cardiac contractility in heart failure after myocardial infarction, demonstrating the possible role of O-GlcNAcylation in the development of chronic heart dysfunction [56]. Increased O-GlcNAcylation induces hypertrophy-like changes in mouse hearts [140]. However, the exercise-induced development of physiological hypertrophy in mice is associated with reduced protein O-GlcNAcylation by improving contractile function [46].

OGA activity is strongly associated with heart failure, and decreased O-GlcNAcylation is beneficial for combating pressure-overload-induced pathological remodeling and heart failure. Muthusamy et al. reported a decrease in OGA expression in heart failure, and increased O-GlcNAcylation has a proadaptive effect in mouse heart failure [141]. OGT transgenic hearts exhibit an increase in O-GlcNAcylation and progress to severe dilated cardiomyopathy, ventricular arrhythmias and premature death. Mitochondrial energy disorders and impaired complex I activity were observed in the hearts of OGT transgenic mice. The transgenic expression of OGA rescued the activity of complex I, indicating that mitochondrial complex I plays an important role in O-GlcNAcylation-mediated cardiac pathology [22]. Dassanayaka et al. reported that altering the ability of myocardial cells to add or remove O-GlcNAcylation to proteins could exacerbate early infarct-induced heart failure in mice [58]. The increase in protein O-GlcNAcylation for 24 weeks leads to mice cardiac hypertrophy, mitochondrial dysfunction, fibrosis and diastolic dysfunction. An increase in O-GlcNAcylation in the heart can lead to impaired calcium dynamics and reduced VDAC and CaMKII activation, which induces arrhythmias, mitochondrial dysfunction and maladaptive hypertrophy, microvascular dysfunction, fibrosis and cardiomyopathy. These harmful effects can be prevented by inhibiting O-GlcNAcylation through the upregulation of AMPK and SIRT1, or by inhibiting OGT or stimulating OGA [142]. OGT deficiency in mice leads to severe postnatal death, and the surviving mice have dilated hearts and observable signs of heart failure with reduced expression of contractile genes [122]. However, restoring OGA expression partially rescued the dilation and dysfunction of the heart by normalizing the O-GlcNAcylation level of cardiac protein [143].

Previously, studies have shown that augmented O-GlcNAcylation may promote hypertrophy, mitochondrial autophagy and fibrosis to exacerbate right ventricular dysfunction and remodeling induced by intermittent hypoxia for 4 weeks in mice, and these effects are mediated by GSK-3β inactivation, an increased p-Drp-1/MFN2 ratio and Smad2 activation [55]. Further research revealed that OGT transgenic mice exhibit a significant decrease in cardiac function and thinning of the left ventricle myocardium with cardiac systolic dysfunction four weeks after TAC surgery. Moreover, the O-GlcNACylation of GSK-3β at Ser9 was increased, and the O-GlcNAcylation of GSK-3β caused compensatory cardiac hypertrophy to mediate increased O-GlcNAcylation-aggravated pressure-overload-induced heart failure [26]. The PGC-1α as a key factor in regulating metabolism is crucial for normal cardiac function. Its activity is inhibited during POH. Research has found that the inhibitory effect of PGC-1α on mouse myocardial hypertrophy is at least partially regulated by O-GlcNAc signaling, and that PGC-1 is O-GlcNAcylated. The reduction in O-GlcNAc signaling alleviates the expression of PGC-1α and most of its downstream genes. Enhancing O-GlcNAc signaling can also reduce glucose starvation-induced PGC-1α upregulation [35]. AMP-activated protein kinase (AMPK) has been shown to inhibit myocardial hypertrophy. AMPK mainly mediates GFAT phosphorylation to reduce the O-GlcNAcylation of proteins such as troponin T and subsequently prevent cardiac hypertrophy predominantly; decreasing the O-GlcNAcylation by GFAT inhibitors of glutamine blocks cardiomyocyte hypertrophy, mimicking AMPK activation [41].

### 4.2. Diabetic Cardiomyopathy

Diabetes-induced hyperglycemia is the central driving factor of the diabetic heart response. This response includes cardiomyocyte hypertrophy, fibrosis and oxidative stress and is known as diabetic cardiomyopathy. Diabetic cardiomyopathy is one of the most common complications of diabetes and is characterized by diastolic dysfunction and left ventricular hypertrophy [144]. Diabetic cardiomyopathy is a serious consequence of diabetes and is related to the increased risk of coronary artery disease, heart failure and arrhythmia [144,145]. Studies have reported that the expression of left ventricular HBP markers (OGT, OGA and GFAT1/2) and total O-GlcNAcylation were increased in diabetes [135,146,147,148]. The O-GlcNAcylation of nuclear and cytoplasmic proteins is involved in the pathology of diabetes complications including cardiomyopathy [149,150]. In the human myocardium, total protein O-GlcNAcylation is increased in diabetic patients compared with non-diabetic patients and is related to left ventricular dysfunction [135]. In human diabetic hearts, O-GlcNAc levels are increased and OGT and OGA delocalized. And in diabetic rat hearts, the specific removal of O-GlcNAcylation restores the myofilament response to Ca^2+^ in diabetic hearts which is due to the subcellular redistribution of OGT and OGA [151]. Mice with type 2 diabetes (T2D) exhibit an increase in the protein O-GlcNAcylation of cardiac endothelial cells. OGA overexpression in endothelial cells significantly reduced O-GlcNAcylation in cardiac endothelial cells, increased the CFVR and capillary density and reduced endothelial cell apoptosis in mice with T2D [72]. A study showed that rat cardiac fibroblasts cultured in high-glucose conditions had increased overall protein O-GlcNAcylation. The nuclear transcription factors Sp1 and arginase II support the occurrence of excessive O-GlcNAcylation in high-glucose cells [152]. The mass spectrogram of O-GlcNAcylation in the heart mitochondria of STZ-induced diabetic rats showed that 86 mitochondrial proteins were O-GlcNAcylated. Many of them exhibited site-specific changes in O-GlcNAcylation in response to diabetes, which indicates that protein O-GlcNAcylation is a new regulatory function that mediates adaptive changes in the mitochondrial metabolism of diabetes cardiomyopathy [78]. In STZ-induced diabetic rats, the mitochondrial OGT was increased with decreased mito-specific OGA [79].

Diabetes has also been associated with the dysregulation of autophagy. Research has shown that O-GlcNAcylation and autophagy are associated with myocardial injury in diabetic rats [112]. Immunoprecipitation studies revealed that Beclin-1 and the antiapoptotic protein Bcl-2 are targets of O-GlcNAcylation. Eight weeks after STZ-induced type 1 diabetic rats, the O-GlcNAcylation of the autophagy markers LC3II/I and P62 significantly increased. The selective OGA inhibitor thiamet G increases O-GlcNAcylation to further disrupt autophagic flow and deteriorate cardiac diastolic function. In contrast, O-GlcNAc antagonists reduce the level of O-GlcNAcylation to maintain autophagic flow and improve cardiac diastolic function [153].

It has been reported that ROS and O-GlcNAcylation are increased in diabetes patients and diabetic mice cardiac, and both ROS and O-GlcNAcylation can activate CaMKII autonomously [134,154]. However, the main mechanisms of diabetes type and disease stage might be different. ROS and O-GlcNAcylation may also differentially regulate cardiac ion channels in diabetes and hyperglycemia [134]. Hyperglycemia in diabetes is related to an increased risk of cardiac dysfunction and arrhythmia, and involves CaMKII function and ROS [155]. Hyperglycemia acutely increases ROS production through O-GlcNAcylation and CaMKII activation. O-GlcNAcylation of CaMKII-S280 can downregulate K^+^ channel expression and function and further increase the susceptibility to arrhythmia in diabetes complicated with hyperglycemia; O-GlcNAcylation may limit the reuptake of Ca^2+^ from the sarcoplasmic reticulum, leading to impaired excitation–contraction coupling and arrhythmia in patients (human right atrial appendage and left ventricular tissue samples) with diabetes complicated with hyperglycemia [6,129,156]. The myocardial sarcoplasmic reticulum Ca^2+^-ATPase (SERCA2a) plays an important role in rat cardiac myocardial contraction. Phospholamban regulates the function of SERCA2a, and phospholamban is O-GlcNAcylated to inhibit phosphorylation, which is related to the deterioration of cardiac function [157]. The ginsenoside Rb1 can significantly ameliorate cardiac dysfunction and abnormal calcium signaling in cardiomyocytes in diabetes by reducing Ca^2+^ leakage, increasing the uptake of Ca^2+^ by SERCA 2a and eliminating the O-GlcNAacylation of calcium processing protein [158].

It is worth noting that the increase in total cardiac O-GlcNAcylation is a mechanism by which exercise is beneficial to type 2 diabetic mice hearts [159]. Research has shown that the excessive O-GlcNAcylation of the cardioprotective enzyme acetaldehyde dehydrogenase 2 (ALDH2) is the mechanism by which hyperglycemia exacerbates myocardial I/R injury of rats. Alda-1, a specific activator of ALDH2, can significantly reduce ALDH2-O-GlcNAcylation to improve I/R combined with hyperglycemia-induced infarct size, the cell apoptosis index and cardiac dysfunction [160]. Kronlage et al. further reported that the O-GlcNAcylation of HDAC4 at Ser-642 has a cardioprotective effect on mice diabetes and counteracts pathological CaMKII signal transduction [161]. Selective targeting of the O-GlcNAcylation of cardiac proteins to restore the physiological O-GlcNAcylation balance may constitute a new therapeutic method for diabetes-induced heart failure. Therefore, targeting excessive O-GlcNAcylation or specific target proteins in the heart is a potential therapeutic option for the treatment of cardiac glycotoxicity in diabetes.

### 4.3. Myocardial Ischaemia/Reperfusion (MI/R) Injury

Before MI/R, acute hyperglycemia in the entire rat heart causes myocardial oxidative stress, cell apoptosis and cardiac contractile dysfunction. Elevated glucose levels can cause unstable electrical activity, the disruption of Ca^2+^ homeostasis and a decreased rat myocardial cell survival rate [162]. Myocardial glucose metabolism is associated with O-GlcNAcylation, which is related to cellular stress and death. The increased O-GlcNAcylation in the cardiovascular system is explained as a potential self-protection alarm signal or pressure response in the I/R injury of human isolated atrial trabeculae [37]. Many studies have shown that in both in vitro and in vivo I/R models, an increase in O-GlcNAc level during hypoxia or reperfusion can improve functional recovery and exert anti-ischemic myocardial protective effects through various mechanisms [38,126,162]. Enhanced O-GlcNAcylation can indeed increase survival by activating endogenous stress responses under acute stress conditions, including hypoxia and local ischemia. Thus, it can alleviate calcium overload and oxidative stress [97], cause the formation of mPTPs, enhance mitochondrial Bcl-2 translocation [110,111,113,163,164], reduce endoplasmic reticulum ER stress [47] and activate antiapoptotic pathways [38,128,165,166].

The transcriptional factor XBP1s can upregulate the expression of specific enzymes related to HBP and subsequently promote O-GlcNAcylation. The acute stimulation of XBP1s in the heart by I/R partially induces HBP to confer strong cardioprotective effects, mediated by increased protein O-GlcNAcylation level [167,168,169]. New research has shown that the overexpression of transcript induced in spermiogenesis 40 can alleviate MI/R injury of in male mice by enhancing HBP flux and protein O-GlcNAcylation [24]. However, Wang et al. reported that the upregulation of O-GlcNAcylation induced by hyperglycemia and hyperinsulinemia in diabetic mouse hearts leads to a low survival rate and increased infarct size in diabetic MI/R [170]. Moreover, hyperglycemia weakens the protective effect of insulin against MI/R injury through hyperglycemia-induced O-GlcNAcylation and inactivation of insulin signaling proteins in mice or dogs [40,128]. These findings indicate that the rapidly increasing O-GlcNAcylation is an adaptive cardiac response that can temporarily protect myocardial function. If the O-GlcNAcylation level is maintained at a high level in chronic diseases or for a long time, it is harmful to myocardial cells.

O-GlcNAcylation seems to depend not only on the circulating glucose concentration, but also on the cardioprotective effect of ischemic preconditioning (IPC) [171]. IPC reduces infarct size by increasing the levels of O-GlcNAc and OGT levels, and OGA activity in I/R-injured isolated rat hearts [172]. The increased levels of myocardial glucose uptake and O-GlcNAc are involved in the mechanisms of IPC [25,173]. Study indicated that salidroside significantly enhances glucose uptake and protein O-GlcNAacylation levels to alleviate myocardial cell damage after I/R [174]. Postconditioning with sevoflurane can inhibit rats MI/R-induced necrosis by regulating OGT-mediated O-GlcNAcylation of RIPK3 and reducing the formation of RIPK3/MLKL complexes [42]. And isoflurane induces O-GlcNAcylation of mitochondrial VDAC to inhibit the opening of mPTPs and confer resistance to I/R stress [45].

### 4.4. Atherosclerosis and Coronary Heart Disease (CHD)

Atherosclerosis is a key predisposing factor in the development of CHD and leads to much morbidity and mortality worldwide, including most myocardial infarction and many strokes, as well as disabling peripheral artery disease [23,175,176]. Several pre-disposing factors, including low-density lipoprotein levels, arterial protein networks, hypertension, cigarette smoking and diabetes mellitus, have been identified [177,178,179]. Accelerated atherosclerosis is the main cause of morbidity and mortality in diabetes. Hyperglycemia is considered as an independent risk factor for the aggravation of atherosclerosis in diabetes. In diabetic ApoE-null mice, high glucose/hyperglycemia regulates vascular A20 expression through O-GlcNAcylation-dependent ubiquitination and proteasome degradation [180]; O-GlcNAcylation of A20 has a crucial role in the negative regulation of NF-κB signaling cascades to protect against inflammation-induced vascular injury [181]. An acute increase in protein O-GlcNAcylation prevents TNF-α-induced vascular dysfunction by inhibiting iNOS expression [36]. Under the conditions of diabetes and high glucose, eNOS undergoes heavy O-GlcNAcylation, which inhibits its phosphorylation and production of nitric oxide (NO) to restore the activity of eNOS and prevent human umbilical vein endothelial cell dysfunction in CVDs [182].

Research has reported that OGA overexpression can inhibit p53 or downregulate p53 to reduce coronary endothelial cell apoptosis and improve cardiac function in diabetic mice [183]. Endothelin-1 enhances the level of protein O-GlcNAcylation in blood vessels, which contributes to an increase in the vascular constriction response by activating the RhoA/Rho kinase pathway in rat vascular smooth muscle cells [184]. The elevated protein O-GlcNAcylation in hyperglycemia is related to atherosclerotic lesion formation and the increased proliferation of vascular smooth muscle cells [185]. A recent study revealed that smooth muscle OGT (smOGT) plays a direct role in the formation of atherosclerosis induced by hyperglycemia and the dedifferentiation of smooth muscle cells [124].

### 4.5. Hypertension and Arrhythmia

Hypertension is one of the most common chronic diseases in the world and is a key risk factor for the development of other CVDs. In recent years, research has emphasized the potential role of O-GlcNAcylation in the pathogenesis of hypertension [23,132]. Studies have shown that O-GlcNAcylation regulates vascular reactivity under normal glucose conditions and that the vascular O-GlcNAcylation of rats is increased in DOCA-salt-induced hypertension. Regulating the increase in vascular O-GlcNAcylation may be a new treatment method for mineralocorticoid-induced hypertension [186].

In myocardial cells, an increased glucose concentration significantly increases the CaMKII-dependent activation of spontaneous sarcoplasmic reticulum Ca^2+^ release events, which may lead to cardiac mechanical dysfunction and arrhythmias. The inhibition of O-GlcNAc signaling or ablation of the CaMKIIδ gene could prevent these effects. In a fully perfused heart, an increase in glucose concentration through O-GlcNAc and CaMKII-dependent pathways exacerbates arrhythmias. In human, rat and mouse, Erickson JR et al. identified a novel mechanism linking CaMKII and hyperglycemic signaling in diabetes mellitus, which is a key risk factor for heart and neurodegenerative diseases. In diabetic animals, the acute blockade of O-GlcNAc prevents the occurrence of arrhythmia [187]. In a rat model of diabetes, diabetic cardiomyopathy can increase the risk of fatal ventricular arrhythmia. Hyperglycemia increases the O-GlcNAcylation of Nav1.5 and reduces the interaction between Nav1.5 and Nedd4-2/SAP-97, leading to the abnormal expression and distribution of Nav1.5, loss of sodium channel function and prolongation of the PR/QT interval. Excessive O-GlcNAcylation of Nav1.5 is a newly identified signaling event that may be a potential factor in cardiac arrhythmia in diabetes [188]. Another study showed that a selective sodium glucose cotransporter-2 inhibitor weakens the inducibility of ventricular arrhythmias by normalizing intracellular Ca^2+^ processing, at least partially through the inhibition of O-GlcNAcylation via the inhibition of glucose uptake into mice cardiomyocytes [125].

Therefore, understanding the regulation of O-GlcNAcylation in specific cardiovascular cells and subcellular compartments under physiological and pathological conditions is key to revealing the mechanism by which O-GlcNAcylation regulates the pathogenesis of CVDs (Table 2 and Figure 3).

## 5. Conclusions and Future Perspectives

O-GlcNAcylation is widely and dynamically present in mitochondria. Mitochondrial proteins are the main targets of O-GlcNAcylation. The O-GlcNAcylation of mitochondrial proteins plays an important role in mitochondrial dynamics, mitochondrial biosynthesis, mitochondrial oxidative stress, mitophagy, mitochondrial apoptosis and mitochondrial energy metabolism, which can protect or weaken mitochondrial function. The O-GlcNAcylation of the mitochondrial proteome may regulate mitochondrial energy metabolism by regulating the TCA and the composition of the mitochondrial respiratory chain, and it also involves the threshold of calcium-induced permeability transition pore opening, mitochondrial dynamics and migration rate, mitochondrial morphology, acetylation of the mitochondrial proteome and mitochondrial oxidative stress. Therefore, O-GlcNAcylation plays a unique role in regulating mitochondrial homeostasis.

Notably, O-GlcNAcylation plays a crucial role in regulating cardiovascular function. The O-GlcNAcylation of cardiovascular proteins is a dynamic process that is crucial for maintaining normal myocardial cell function. In healthy myocardial cells, due to changes in nutrient availability, in response to different physiological stimuli, the overall O-GlcNAc level of the cells changes within the optimal range within minutes, hours or longer. The levels of these proteins and external pressure sources may cause changes in O-GlcNAcylation levels beyond the optimal range, either too high or too low, which may lead to cellular dysfunction and potential cell death, depending on the duration of these deviations. Changes in O-GlcNAcylation play important roles in heart remodeling, heart failure, diabetes cardiomyopathy, ischemic heart disease, hypertension and arrhythmia.

The heart is the organ with the richest mitochondrial content in the body. Mitochondria are the main organelles that regulate heart function and myocardial cell viability under physiological and pathological conditions, and they are the most sensitive organelles to various injuries. Thus, regulating the modification of mitochondrial protein O-GlcNAc and maintaining mitochondrial homeostasis may be new strategies and targets for the early prevention and treatment of CVDs.

Importantly, the imbalance of O-GlcNAcylation can lead to mitochondrial dysfunction. There are pharmacological interventions of O-GlcNAcylation activity that have been shown to enhance mitochondrial function. The inhibition or overexpression of OGT or OGA activity (including glucosamine, TMG, OSMI-1, PUGNAc) significantly affects energy production, the MMP, apoptosis, mitochondrial oxygen consumption with decreased loss of ΔΨ(m), inhibited intracellular ROS and mitochondrial calcium overload. These results provide a more comprehensive overview of potential therapeutic strategies for addressing mitochondrial dysfunction in the context of cardiac hypertrophy and heart failure. 

## Figures and Tables

**Figure 1 antioxidants-13-00571-f001:**
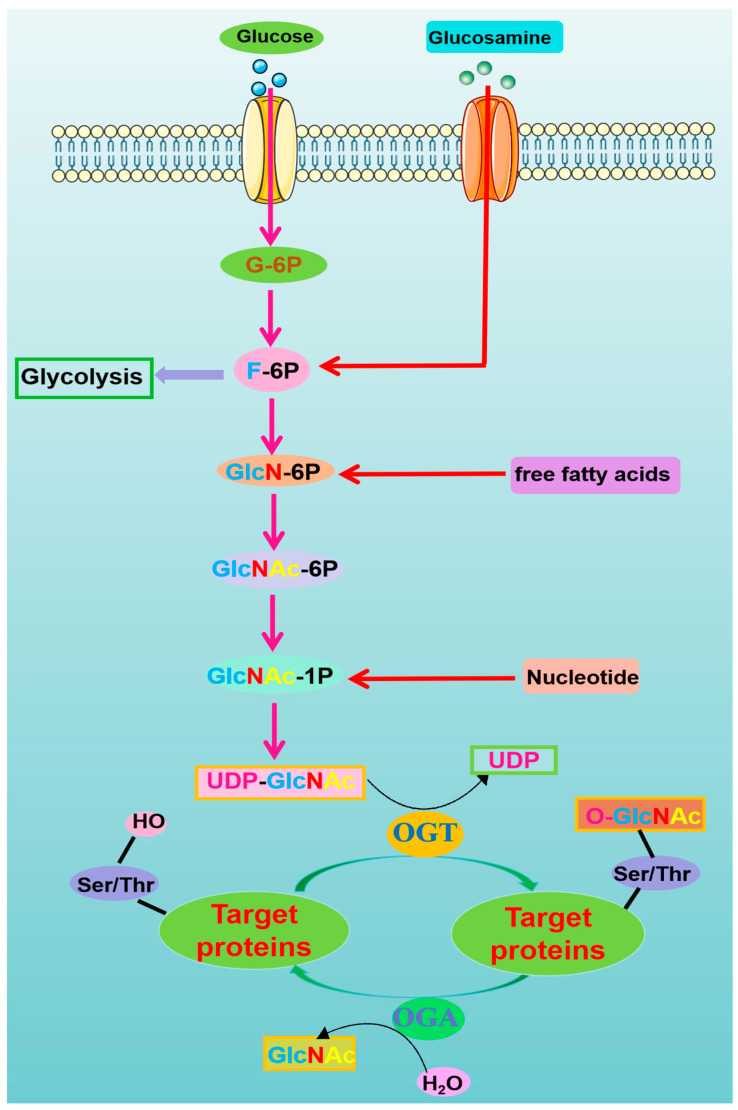
Production of O-GlcNAc and regulation of protein O-GlcNACylation. G-6P, Glucose-6-phosphate, F-6P, fructose-6-phosphate; OGT, O-GlcNAc transferase; OGA, O-GlcNAcase; Ser, serine; Thr, threonine; OH, hydroxyl group.

**Figure 2 antioxidants-13-00571-f002:**
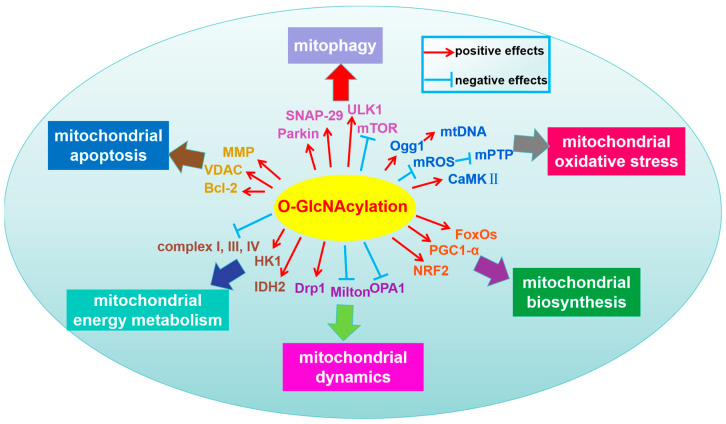
O-GlcNAcylation regulates mitochondrial homeostasis under normal and stressed conditions. Mitochondrial dynamics: An increased level of O-GlcNAcylation could upregulate the level of Drp1 and induce the translocation of Drp1 from the cytoplasm to the mitochondria to affect the balance of mitochondrial fusion and fission. Milton is an essential substrate for OGT that prevents mitochondrial motility by locating and mutating key O-GlcNAcylated serine residues. High-glucose conditions led to a decreased OPA1 protein level with increased O-GlcNAcylation, which led to mitochondrial dysfunction by increasing mitochondrial fragmentation. The balance of mitochondrial fission and fusion is regulated by the O-GlcNAcylation of GTPases in response to cellular stimulation. Mitochondrial biosynthesis: PGC1-α binds to OGT and targets the enzyme to FoxOs, leading to an increase in its O-GlcNAcylation and transcriptional activity. O-GlcNAcylation signaling increased the protein expression levels of NRF2 to regulate mitochondrial function. Mitochondrial oxidative stress: Increased O-GlcNAcylation decreased mROS levels, thereby regulating ROS homeostasis and attenuating the formation of mPTP and the subsequent loss of MMP to induce mitochondrial dysfunction. Hyperglycemia-induced O-GlcNAcylation of Ogg1 mediated increased mtDNA damage and mitochondrial oxidative stress damage. Acute increase in the extracellular glucose concentration leads to the O-GlcNAcylation of CaMKII, which results in the release of Ca^2+^ from the SR and induces ROS production to impair calcium dynamics and contractile derangements, as well as mitochondrial dysfunction. Mitophagy: Mitochondrial homeostasis was affected by O-GlcNAcylation through PINK1-dependent mitophagy. O-GlcNAcylation can reduce mTOR activation to promote mTOR-dependent autophagy. The O-GlcNAcylation of ULK1 and SNAP-29 mediate autophagy to maintain mitochondrial homeostasis. Mitochondrial apoptosis: Mitochondria with elevated O-GlcNAc levels have high levels of VDAC O-GlcNAcylation and are more resistant to calcium-induced swelling. The enhancement of O-GlcNAc levels attenuated the loss of MMP by increasing mitochondrial Bcl-2 levels to decrease apoptosis. Mitochondrial energy metabolism: Members of mitochondrial respiratory chain complexes, such as the NDUFA9 subunit of complex I, the core 1 and core 2 subunits of complex III and the mtDNA-encoded subunit I of complex IV, are O-GlcNAcylated. Increased mitochondrial O-GlcNAcylation is associated with the impaired activity of complexes I, III and IV, as well as decreased mitochondrial calcium and cellular ATP content. The dynamic O-GlcNAacylation of HK1 promotes the assembly of the glycolytic metabolites on the outer membrane of mitochondria and mitochondrial ATP production. O-GlcNAcylation enhanced the protein half-time of IDH2 by inhibiting ubiquitin-mediated proteasome degradation. The positive effects of proteins’ O-GlcNAcylation are indicated by red arrows, and the negative effects of proteins’ O-GlcNAcylation are shown with blue arrows. Drp1, dynamic-related protein 1; OPA1,optic atrophy 1; NRF2, nuclear factor erythroid 2-related factor 2; PGC1-α, Peroxisome proliferator-activated receptor gamma coactivator-1 alpha; FoxOs, Forkhead box protein O s; CaMKII, calmodulin-dependent protein kinase II; mROS, mitochondrial reactive oxygen species; Ogg1, 8-oxoguanine DNA glycosylase; mPTP, mitochondrial permeability transition pore; mtDNA, mitochondrial DNA; SNAP-29, synaptosomal-associated protein 29; ULK1, serine/threonine-protein kinase ULK1; mTOR, mechanistic target of rapamycin; MMP, mitochondrial membrane potential; VDAC, voltage-dependent anion channel; Bcl-2, B-cell lymphoma-2; HK1, hexokinase 1; IDH2, isocitrate dehydrogenase 2.

**Figure 3 antioxidants-13-00571-f003:**
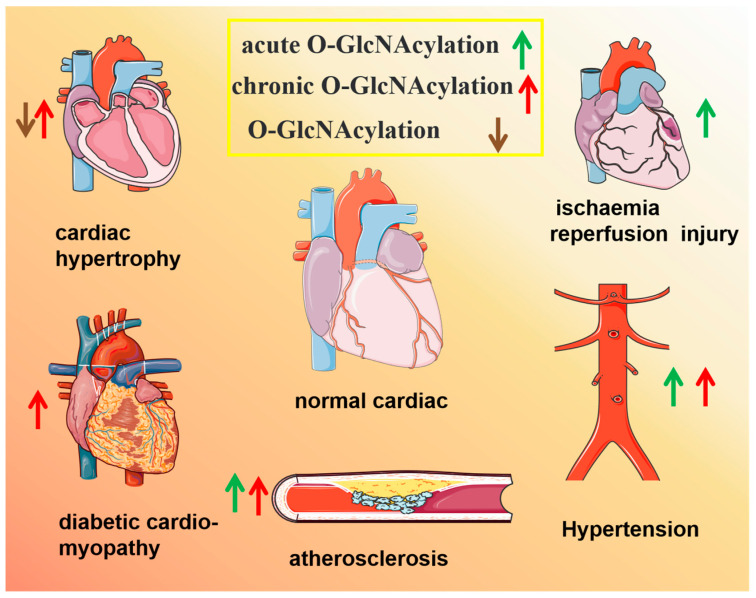
Roles and pathological effects of protein O-GlcNAcylation in CVDs. Pressure-overload hypertrophy augments HBP flux to increase the O-GlcNAcylation level. Heart failure has been associated with an increase in the O-GlcNAcylation rate, which induces cardiomyopathy. Decreased O-GlcNAcylation is beneficial for combating pressure-overload-induced pathological remodeling and heart failure. Total protein O-GlcNAcylation is increased in diabetes and is related to left ventricular dysfunction. The increased O-GlcNAcylation is explained as a potential self-protection alarm signal or pressure response in ischemia/reperfusion injury. However, the upregulation of O-GlcNAcylation induced by hyperglycemia and hyperinsulinemia in diabetic mouse hearts leads to a low survival rate and increased infarct size in diabetic MI/R. An acute increase in protein O-GlcNAcylation prevents TNF-α-induced vascular dysfunction by inhibiting iNOS expression. Under the conditions of diabetes and high glucose, the chronic increase in O-GlcNAcylation induces vascular injury. O-GlcNAcylation regulates vascular reactivity under normal glucose conditions and vascular O-GlcNAcylation is increased in DOCA-salt-induced hypertension. Increased glucose concentration significantly increases the O-GlcNAcylation level which may lead to cardiac mechanical dysfunction and arrhythmias. In diabetic animals, the acute blockade of O-GlcNAc prevents the occurrence of arrhythmia. Acute O-GlcNAcylation elevation is indicated as a positive effect by green arrows; chronic O-GlcNAcylation elevation is shown with red arrows as a negative effect; The decrease in the O-GlcNAcylation level is indicated by gray arrows.

**Table 1 antioxidants-13-00571-t001:** List of O-GlcNAcylation proteins and effects of mitochondrial homeostasis.

Target Protein	Regulatory Effects on Protein	Downstream Effects	Mitochondrial Homeostasis	Refs.
Drp1	upregulates Drp1 level and activity	increases mitochondrial fission	mitochondrial fusion and fission	[73,83]
Milton	enhances enzyme activity	reduces mitochondrial motility	mitochondrial dynamics	[82]
OPA1	decreases OPA1 protein level	increases mitochondrial fragmentation, reduces the MMP and the activity of mitochondrial complex IV	mitochondrial dysfunction	[84]
PGC1-α	increases the activity of PGC1-α and the O-GlcNAcylation of specific transcription factors such as FoxOs	mediates gluconeogenesis	mitochondrial biogenesis and mitochondrial density	[86,87,88,89]
NRF2	increases protein activation	decreases ROS levels to antioxidant response and enhance respiration	mitochondrial biosynthesis	[5,91]
mROS	reduces ROS generation	attenuates the formation of mPTPs and the subsequent loss of MMP	mitochondrial oxidative stress, apoptosis, respiration	[5,9,50,93,96,97,98]
Ogg1	increases protein activition	leads to mtDNA damage and oxidative stress	mitochondrial oxidative stress damage	[94]
CaMKII	increases protein activation	impaired calcium dynamics and contractile derangements; mitochondrial calcium overload	mitochondrial dysfunction	[6,44,98]
mTOR	suppresses proteasome activity	mitophagy	maintains mitochondrial fitness and enhances autophagy flux	[12]
Parkin	increases protein activation	mitophagy	mitochondrial quality and mitochondrial homeostasis	[101,102,103]
ULK	increases protein activation	mitpphagy	mitochondrial homeostasis	[104]
SNAP-29	increases protein activation	mittophagy	mitochondrial homeostasis	[106,110]
MMP	MMP upregulation	inhibits cell apoptosis	mitochondrial apoptosis	[5]
VDAC	increases protein activation	reduces mitochondria-related apoptosis	mitochondrial apoptosis	[52,108,109]
mPTP	restrains mPTP formation	reduces oxidative stress	mitochondrial apoptosis	[70,98,111],
Bcl-2	increases mitochondrial Bcl-2 levels	attenuates the loss of MMP	mitochondrial apoptosis	[112,113]
ETC complexes complex I, complex III, complex IV	impairs activity of complexes I, III and IV	urea cycle, TCA cycle and lipid metabolism	mitochondrial dysfunction and impaired energy function of mitochondria	[8,63,69,73,115,116,117,118]
HK1	enhances the mitochondrial binding of HK1	coordinates glycolysis and mitochondrial ATP production	Mitochondrial energy metabolism	[10]
IDH2	enhances the protein half-time of IDH2	increases glycolysis and TCA cycle metabolites	Mitochondrial energy metabolism	[119]

Drp1, dynamic-related protein 1; OPA1, optic atrophy 1; NRF2, nuclear factor erythroid 2-related factor 2; PGC1-α, Peroxisome proliferator-activated receptor gamma coactivator-1 alpha; FoxOs, Forkhead box protein O s; CaMKII, calmodulin-dependent protein kinase II; mROS, mitochondrial reactive oxygen species; Ogg1, 8-oxoguanine DNA glycosylase; mPTP, mitochondrial permeability transition pore; mtDNA, mitochondrial DNA; SNAP-29, synaptosomal-associated protein 29; ULK1, serine/threonine-protein kinase ULK1; mTOR, mechanistic target of rapamycin; MMP, mitochondrial membrane potential; VDAC, voltage-dependent anion channel; Bcl-2, B-cell lymphoma-2; HK1, hexokinase 1; IDH2, isocitrate dehydrogenase 2; TCA, tricarboxylic acid cycle; ETC, electron transport chains.

**Table 2 antioxidants-13-00571-t002:** Effects of O-GlcNAcylation in cardiovascular diseases (CVDs).

O-GlcNAcylation Levels		Effect	Diseases	Refs.
normalized level		positive	cardiac hypertrophy and heart failure	[143]
elevated level	acute increased	negative	hypertension	[186]
		protective	MI/R	[25,37,38,42,45,47,97,110,111,112,113,126,128,162,163,164,165,166,167,168,169,171,172,173,174]
			atherosclerosis	[36,184]
	chronic increased	negative	cardiac hypertrophy and heart failure	[14,26,35,44,57,121,137,138,139,140,142]
			diabetic cardiomyopathy	[6,78,79,129,134,135,146,147,148,149,150,151,152,154,155,156,157]
			diabetic MI/R	[40,170]
			atherosclerosis and coronary heart disease	[124,180,182,185]
			hypertension, arrhythmia with diabetes	[125,187,188]
decreased level		protective	cardiac hypertrophy and heart failure	[41,46]
			diabetic cardiomyopathy	[153,158,160]
			coronary heart disease with diabetes	[124,183,185]

## Data Availability

Not applicable.

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
