# Peer review of "Roles of O-GlcNAcylation in Mitochondrial Homeostasis and Cardiovascular Diseases"

_antioxidants, 2024, doi:10.3390/antiox13050571_

Round 1
Reviewer 1 Report
Do you have any information about studies regarding this process to human? In many points you discuss about results on studies without specification if studies are on cells on enzyme?
Fig 2 - I recommend to put use different color depending on the positive or negative effects o O-GlcNAcylation
Reviewer 2 Report
This review article provided a comprehensive overview of the mitochondrial homeostasis regulated by protein O-GlcNAcylation, and the key roles of O-GlcNAcylation in the pathogenesis of cardiovascular diseases (CVDs), including heart failure, cardiac hypertrophy, myocardial infarction (MI), and myocardial ischemia/reperfusion (MI/R)injury.
The reviewer considers that the authors have well written the present review article, and has some comments as follows:
Major comments:
1. Section 4.1 on Cardiac Hypertrophy and Heart Failure: The manuscript discusses heart failure, which encompasses both systolic and diastolic dysfunction. However, it lacks specificity regarding which type of heart failure is addressed. For clarity and precision, it is imperative that the authors explicitly state whether the discussion pertains to systolic dysfunction, diastolic dysfunction, or both, with regard to both the organ level and the myocyte level.
2. In relation to the discussion on mitochondrial dysfunction, if there are pharmacological interventions available that have been shown to enhance mitochondrial function, it is essential that these be included in the discussion. This addition would provide a more comprehensive overview of potential therapeutic strategies for addressing mitochondrial dysfunction in the context of cardiac hypertrophy and heart failure.
Minor comment:
3. The manuscript contains sections that are formatted inconsistently, with text appearing in various colors (such as gray or red) and different font types. These formatting issues should be addressed to ensure uniformity and readability throughout the document.
Reviewer 3 Report
Although interesting, this manuscript needs to be improved before publication.
1. Make it clear that multiple nutrients contribute to both UDP-GlcNAc and protein O-GlcNAc synthesis. Although glucose availability is an important factor in O-GlcNAc synthesis, glutamine is critical as the amine donor for glucosamine 6-phosphate, while fatty acid metabolism is probably the main source for the acetyl moiety.
2. Figure 2 legend. Please, complete the meaning of all acronyms. Please, indicate the effect of O-GlcNAcylation on the described processes (in figure and legend).
3. Line 603. It is not clear whether the sentence “O-GlcNAcylation has been implicated in the adverse effects of diabetes on the heart” is a new section or not.
4. In the review, it is not clear in which situations O-GlcNAcylation has negative or positive effects. It would be of interest to summarize these effects on a table.
5. Figure 3. The figure 3 legend should be more explanatory, particularly about the positive or negative effect of O-GlcNAcylation.
6. It would be of interest to include a section on the modulation of O-GlcNAcylation and its therapeutic implications.
7. It would be of interest to discuss the role of O-GlcNAcylation in epigenetics and how this regulation may affect mitochondrial function.
Round 2
Reviewer 2 Report
The reviewer has no further comment.
The reviewer has no further comment.
Reviewer 3 Report
The authors have adressed all my concerns
The authors have adressed all my concerns